# Pathway Analysis of Fucoidan Activity Using a Yeast Gene Deletion Library Screen

**DOI:** 10.3390/md17010054

**Published:** 2019-01-14

**Authors:** Monika Corban, Mark Ambrose, Joanne Pagnon, Damien Stringer, Sam Karpiniec, Ahyoung Park, Raj Eri, J Helen Fitton, Nuri Gueven

**Affiliations:** 1School of Medicine, University of Tasmania; Hobart TAS 7001, Australia; Monika.Corban@utas.edu.au (M.C.); Mark.Ambrose@utas.edu.au (M.A.); joanne.Pagnon@utas.edu.au (J.P.); 2Marinova Pty Ltd., Cambridge TAS 7170, Australia; damien.stringer@marinova.com.au (D.S.); sam.karpiniec@marinova.com.au (S.K.); ahyoung.park@marinova.com.au (A.P.); 3School of Health Sciences, University of Tasmania, Newnham TAS 7248, Australia; raj.eri@utas.edu.au

**Keywords:** fucoidan, *Undaria pinnatifida*, yeast gene deletion library, HCT-116

## Abstract

Fucoidan, the sulfated fucose-rich polysaccharide derived from brown macroalgae, was reported to display some anti-cancer effects in in vitro and in vivo models that included apoptosis and cell cycle arrest. The proposed mechanisms of action involve enhanced immune surveillance and direct pro-apoptotic effects via the activation of cell signaling pathways that remain largely uncharacterized. This study aimed to identify cellular pathways influenced by fucoidan using an unbiased genetic approach to generate additional insights into the anti-cancer effects of fucoidan. Drug–gene interactions of *Undaria pinnatifida* fucoidan were assessed by a systematic screen of the entire set of 4,733 halpoid *Saccharomyces*
*cerevsiae* gene deletion strains. Some of the findings were confirmed using cell cycle analysis and DNA damage detection in non-immortalized human dermal fibroblasts and colon cancer cells. The yeast deletion library screen and subsequent pathway and interactome analysis identified global effects of fucoidan on a wide range of eukaryotic cellular processes, including RNA metabolism, protein synthesis, sorting, targeting and transport, carbohydrate metabolism, mitochondrial maintenance, cell cycle regulation, and DNA damage repair-related pathways. Fucoidan also reduced clonogenic survival, induced DNA damage and G1-arrest in colon cancer cells, while these effects were not observed in non-immortalized human fibroblasts. Our results demonstrate global effects of fucoidan in diverse cellular processes in eukaryotic cells and further our understanding about the inhibitory effect of *Undaria pinnatifida* fucoidan on the growth of human cancer cells.

## 1. Introduction

Fucoidans are a class of fucose rich sulfated polysaccharides derived from brown macroalgae that are associated with a large range of bioactivities [1,2]. As a naturally occurring part of edible seaweed, it is already part of the normal diet in many countries. Therefore, certain fucoidan extracts have obtained ‘generally recognized as safe’ (GRAS) status in the US and received novel foods approval in the EU. Fucoidans have long been noted as a selectin blocking compound that inhibits cell–cell interactions [3]. This ability to disrupt cell–cell interactions is likely at least in part, responsible for the potent anti-inflammatory activity of different fucoidan preparations [1]. Fucoidans can also inhibit the adhesion of proteins and organisms to non-biological surfaces, and may help to inhibit biofouling [4]. The anti-microbial activity of fucoidans is also largely based on its inhibition of bacteria-substrate binding [5]. However, in a *C. elegans* infection model, a fucoidan extract increased the immunity of the host organism and downregulated quorum sensing genes in the bacterial pathogen, which suggests that fucoidans also have the potential to impact gene expression and cellular signaling pathways [6]. While fucoidan-mediated effects on yeasts and fungi are largely unexplored, different fucoidan preparations have also been investigated for their anti-cancer activity in vitro and in vivo [7,8]. In vivo, the anti-cancer response appears to be a combination of enhanced immune function, regulation of checkpoint inhibitor levels [9,10], and a direct cytotoxic activity on cancer cells such as DU-145 human prostate cancer cells [11]. In pre-clinical colon cancer cell models, fucoidans induced both apoptosis and cell cycle arrest, while the exact mechanism for this effect remains unclear [12,13,14,15]. One suggested mode of action involves fucoidan-induced endoplasmic reticulum (ER) stress that induces apoptotic cancer cell death via the activation of unfolded protein response (UPR) pathways [14,16,17]. Fucoidan treatment of HCT-116 colon cancer cells resulted in downregulation of the ER protein 29 (ERp29), and activated the phosphorylation of eukaryotic initiation factor 2 alpha (p-eIF2a)/CCAAT/enhancer binding protein homologous protein (CHOP) pro-apoptotic cascade [14]. Surprisingly, another fucoidan preparation was also described to protect against endoplasmic reticulum (ER) stress [18]. Autophagy, necessary for the bulk degradation of cellular components is recognized as an important mechanism for cell survival under conditions of ER stress. In this context, fucoidans are described as antagonists of scavenger receptors and may even protect against or modulate autophagy in macrophages [18,19]. 

Despite a large degree of experimental consistency, the molecular differences in fucoidan preparations significantly complicate the comparison of reported results. To obtain an unbiased view of the multiple, sometimes conflicting, biological activities and signaling mechanisms that are affected by fucoidans in proliferating cells, this study initially examined the effects of a well-defined fucoidan extract from the edible macroalga *Undaria pinnatifida* by screening a *S. cerevisiae* gene deletion library. This eukaryotic model and type of analysis has been used widely in genome-wide phenotypic screens to understand cellular responses to environmental stressors and to deduce drug–gene interactions in higher organisms [20,21,22,23,24]. For this purpose, the present study employed a single-gene deletion library of *S. cerevisiae* strains and incubated the gene deletion strains in the absence and presence of fucoidan. By comparing the overall growth (population density) of the gene deletion strains in the absence and presence of fucoidan we were able to unearth genes, and hence potential genetic/functional pathways impacted by fucoidan. This experimental approach enables a global view of drug–gene interactions in the yeast system, which, due to a high degree of functional conservation, can also inform our understanding of fucoidan-gene interactions in the mammalian system. We used this experimental approach to address the question of how one type of edible fucoidan—from *Undaria pinnatifida*—exerts its reported anti-cancer effects, and what, if any, effects it might have on normal healthy cells. Using this approach for the first time, the present study identified a wide range of fucoidan-affected cellular processes in yeast, some of which were subsequently confirmed in studies involving mammalian cells.

## 2. Results

### 2.1. Yeast Gene Deletion Library Results

To gain an unbiased insight into gene-fucoidan interactions, the growth of the complete library of 4,733 haploid *S*. *cerevisiae* gene deletion strains was measured in the absence and presence of 500 g/mL *Undaria pinnatifida* fucoidan, *UPF*. Based on a cutoff value of 1.5-fold change in cell growth at least 115 deletion strains showed reduced proliferation in the presence of fucoidan compared to control (YPD alone grown) cultures (Appendix A). Out of these, 82 genes (71%) were associated with known cellular processes, while 33 genes (29%) were of unknown function. In contrast, 177 deletion strains showed increased growth in the presence of *UPF* (Appendix A). From these, 136 genes (77%) were associated with well described cellular processes and 41 genes (23%) were of unknown function.

Overall, the data indicated that *UPF* likely interacts with a wide range of genes whose protein are potentially involved in distinct cellular processes, including DNA replication, maintenance and repair, mRNA transcription and processing, ribosome biogenesis, amino acid biosynthesis, carbohydrate and nucleotide metabolism, protein transport and degradation, organelle (mitochondria and vacuole) transport and maintenance, general and oxidative stress responses, and a considerable number of pathways whose precise identities in the eukaryotic/mammalian system remain to be fully determined. To interrogate this dataset in more detail, pathway analysis using String software was employed (Figure 1). In a first iteration, only the 115 genes were assessed whose absence reduced the growth of *S. cerevisiae* in the presence of *UPF* by at least 1.5-fold (Figure 1A). Using a high confidence interaction score of 0.9 (“highest confidence”), the software identified seven major functional groups that included peroxisome biogenesis, amino acid biogenesis, cyclin-cAMP signaling, cell cycle control, DNA repair, RNA polymerase complex, and energy metabolism (Figure 1A).

As a second step, only those genes were interrogated, whose absence increased the growth of *S. cerevisiae* in the presence of *UPF* by at least 1.5-fold (Figure 1B). Using the same stringency settings of 0.9, the software strongly predicted the involvement of the ribosome function and biogenesis, chromatin remodeling and DNA repair, cell cycle checkpoints, mitochondrial stress response, transcription, peroxisome biogenesis, and microtubule-DNA dynamics (Figure 1B).

In a final analysis, all 292 genes identified as being important for the growth of *S. cerevisiae* (by at least 1.5-fold) in the presence of *UPF* were included together in the pathway analysis. This combined analysis revealed some genetic/functional pathways that were not identified in the initial analysis (Figure 1A,B), such as the effects of *UPF* on mitochondrial t-RNA synthase, nucleotide biosynthesis, transcription elongation and splicing, as well as protein transport mechanisms. In addition, this form of analysis strengthened previously identified pathway-*UPF* connections such as an effect on ribosome function, mitochondrial and energy metabolism and DNA repair by adding more genes to previously identified nodes (Figure 1C).

### 2.2. Mammalian Cell Studies

Based on the observed *UPF*-gene interactions in *S. cerevisiae*, a *UPF*-induced cellular stress response was postulated that included cell cycle checkpoints and DNA damage repair pathways. This study intended to translate the genetic pathways identified in yeast to a mammalian system. Since we did not aim to study anti-cancer activity of *UPF* in general, only one well characterized cell line was used exemplary instead of a range of cancer cell lines. To confirm the presence of *UPF*-induced cell cycle control and DNA damage in mammalian cells the general level of toxicity of *UPF* was assessed in HCT-116 colorectal cancer cells. 

#### 2.2.1. Viability Assessment Using WST-1

The WST-1 cell proliferation assay is frequently used as a measure of cell proliferation or cytotoxicity. It is based on the metabolic production of NAD(P)H, which is used as a surrogate marker for cell death, cytostatic activity or metabolic inhibition by test compounds [25]. To assess short-term effects of fucoidan treatment, log-phase HCT-116 colon cancer cells were exposed to *UPF* over 24 h before WST-1 conversion was measured (Figure 2). Surprisingly, the presence of *UPF* at concentrations of up to 100 µg/mL showed no signs of toxicity to the colon cancer cells in the WST-1 assay, regardless of serum concentration in the cell culture media. In fact, low *UPF* concentrations of up to 2.5 µM appeared to significantly increase WST-1 dye conversion (Figure 2). This effect likely represents an artefact of the redox-active nature of the *UPF* preparation as described for other compounds [26], which required us to assess cellular viability in the presence of *UPF* by other assays.

#### 2.2.2. Colony Formation Assay 

In contrast to the WST-1 assay that only detects acute toxicity of a compound based on metabolic activity, the colony formation assay was employed to also assess any long-term effects of *UPF* treatment. Consistent with the WST-1 data, *UPF* showed hardly any effects on the number of colonies after two weeks of continuous treatment. Only the highest *UPF* concentration (100 µM) showed a significant (*p* < 0.05) reduction in colony numbers by about 25% under optimal cell culture conditions (Figure 3A). To assess this effect in more detail, the experiment was repeated but the serum concentration of the cell culture media was reduced to 2%. Since the colony formation assay is essentially measuring ‘proliferative fitness’ of cells in culture, the reduction of growth factors is a commonly used approach to increase the stringency of this assay to detect mild, non-lethal toxicities. While this reduced FCS content did not change the overall cloning efficiency of the cells in the absence of *UPF* (similar numbers of colonies in the untreated cultures at 2% and 10% FCS; data not shown), it had a dramatic effect on the activity of *UPF*. Even at the lowest *UPF* concentration, a highly significant (*p* < 0.001) drop in colony numbers of about 70% was observed (Figure 3A). This toxicity increased further in a dose dependent manner, where only 7% residual colonies were observed at the highest *UPF* concentration (Figure 3A).

To assess if this toxicity was specific to immortalized cells or would also apply to other cell types, the effect of *UPF* was tested on the colony formation activity of non-immortalized primary human dermal fibroblasts (HDF). These cells are significantly more sensitive to their culture conditions and only form colonies with at least 10% fetal calf serum (FCS). Therefore, lower FCS concentrations could not be tested. In contrast to HCT-116 cells, *UPF* treatment of HDF cells led to a bell-shaped concentration curve with a significant increase in colony numbers between 10 and 50 µM. Even at the highest *UPF* concentration (100 µM), HDF still showed at least as many colonies as the untreated control cells (Figure 3B).

#### 2.2.3. Assessment of UPF-induced DNA Damage

The specific toxicity observed in the colorectal cancer cell line was consistent with the yeast data that indicated an interaction between *UPF* and DNA repair pathways. The typical mode of action of most anti-cancer agents is the induction of DNA damage. Nuclear γH_2_AX staining is a marker of DNA damage-induced cellular repair activity and is frequently used as very sensitive marker for the presence of DNA damage, especially DNA double strand breaks [27]. To assess if *UFP* induces DNA damage, non-immortalized human skin fibroblasts (HDF) and colon cancer cells (HCT-116) were exposed to 100 µg/mL *UPF* over 24 h. In untreated HDF, only about 10% of cells showed any γH_2_AX foci at all and these cells displayed typically less than four foci. After exposure to UPF no increase in the number of positive cells was evident and the positive cells also did not show increased numbers of foci (Figure 4A,B) and those HDF with γH_2_AX foci had only very low numbers of foci (Figure 4B). In contrast to HDF, close to 40% of HCT-116 cells showed γH_2_AX foci in the absence of *UPF*, indicative of the genetic instability that is characteristic to many tumor cell lines. More importantly HCT-116 cells showed a very significant (p < 0.01) *UPF*-induced effect with nearly 90% of all cells positive for γH_2_AX foci after 24 h (Figure 4A). In contrast to HDF, HCT-116 cells also showed very high numbers of γH_2_AX foci per cell after exposure to *UPF* (Figure 4C). 

#### 2.2.4. Cell Cycle Assay by Fluorescence-activated Cell Sorting (FACS)

To understand the long-term toxicity of *UPF* on HCT-116 cells in the colony formation assay, cell cycle analysis by flow cytometry was employed. *UPF* induced a significant (*p* < 0.01) G1 arrest at 72 h, while all prior time points remained non-significant. The corresponding reduction in S-phase cells approached significance (*p* = 0.052), while no effects were observed with regards to the number of cells in G2 phase (Figure 5). 

## 3. Discussion

This study initially examined the gene-drug interactions of *Undaria pinnatifida* fucoidan (*UPF*) by screening the complete *S. cerevisiae* gene deletion library to achieve an unbiased genome-wide assessment of the eukaryotic genetic/functional pathways potentially affected by this fucoidan. Especially in cases where (i) a test compound is associated with a multitude of biological effects, or (ii) nothing is known about the biological effects of a test compound, or (iii) a gene of interest, this approach has been used successfully [28]. The present study identified a large number of interacting pathways affected by *UPF* that broadly affected cellular energy metabolism, RNA synthesis, DNA synthesis and repair, cell cycle control, protein synthesis and transport. Our yeast data are in general agreement with a previous report that employed microarray gene expression analysis of *Fucus vesiculosus* extract treated pancreatic cells [29]. While that report also highlighted the effect of fucoidan on DNA damage repair and cell cycle regulation, it did not identify the additional pathways, such as ribosomal or mitochondrial involvement, that our analysis provided for *UPF*.

When the effects of *UPF* on cell cycle control predicted by the yeast results were subsequently tested in a mammalian cell culture system, our results largely confirmed previous reports of G1 arrest. Fucoidan isolated from *Fucus vesiculosus* was previously reported to induce a G1 arrest in the same cell line used in the present study [13], while polysaccharides from peony seeds also induced a G1 arrest in the same cell line [30]. However, in the present study *UPF* led to a much milder G1 arrest that only occurred after 72 hours and with lower magnitude compared to the previous report where a significant increase was already detected after 36 hours [13]. The extent of G1 arrest was much more comparable to the second report, which observed a G1-arrest already after 24 hours, so it remains unclear how many cells would have accumulated in G1 phase after 72 hours [30].

As a major difference to the previous studies [13,30], *UPF* did not induce any overt cytotoxicity or signs of cell death. Previous studies reported polysaccharide-induced loss of membrane integrity, reduced metabolism and apoptosis [13]. In the present study, a distinctive lack of *UPF*-induced cell death or cytotoxicity was evident in the WST-1 and the colony formation assays when the cells were grown under standard conditions. Only when cells were treated under conditions of reduced growth factor availability did *UPF* reduce cellular metabolism. This somewhat hidden sensitivity supports the yeast results that suggest that *UPF* interacts with cellular energy metabolism. Thus, the observed *UPF* hypersensitivity under reduced growth factor signaling indicates a synergistic mode of action. Surprisingly, UPF-induced cytotoxicity at the level of cell proliferation was only observed in the colon cancer cell line but not in non-immortalized human dermal fibroblasts (HDF). Although, it has to be acknowledged that toxicity in these cells could have been present at lower serum concentrations, these cells do not form colonies under these conditions and in contrast to signs of toxicity our results indicate a significant *UPF*-dependent increase in colony formation in HDF. This cell type specific difference was not entirely unexpected, as previous reports highlighted that fucoidans can induce proliferation in primary fibroblasts without inducing toxicity [31,32]. 

To explore the mild toxicity observed in the cancer cell line further, we investigated the possibility of fucoidan-induced DNA damage. Although molecular signatures indicative of fucoidan-induced DNA damage (such as Tunnel staining), have been reported earlier [14], low-molecular weight fucoidan was conversely reported to inhibit DNA-damage induced signaling (including γH_2_AX foci) in etoposide-treated HCT-116 cells [33]. Our results for the first time, indicate a direct and very potent DNA damaging activity by *UPF* by itself that appears to be very selective for the cancer cell line tested. In untreated HCT-116 cells, compared to fibroblasts, elevated basal levels of γH_2_AX foci were detected, which reflects the reported genetic instability and hypermutator phenotype of colorectal cancer cells [34]. Upon exposure to *UPF,* foci numbers in the cancer cell line increased dramatically, to 90 foci/nucleus, which is among the highest numbers reported, while foci numbers in the fibroblasts remained unaffected at a significantly lower level. This significant difference obviously poses the question: how can *UPF* elicit this cell type specific response and how exactly does UPF induce DNA damage? 

Although, the majority of the literature reports anti-oxidant activity of fucoidan, our WST-1 results indicate that *UPF* has some redox activity that, together with the appropriate metabolic background could also act as a pro-oxidant. This double-edged redox activity has been reported for many common antioxidants from vitamins C and E to CoenzymeQ_10_ [35]. There is also evidence that the anti-cancer effects of fucoidans could be attributed to a pro-oxidant activity [36,37]. Whilst *UPF*-induced production of reactive oxygen species (ROS) could well explain the rapid induction of DNA damage observed in the present study, it is striking that despite this toxicity, cell cycle effects only became evident after 72 hours. Even more surprising is the complete lack of detectable cell death in the presence of extensive DNA damage. A different *Undaria pinnatifida* fucoidan extract was previously reported to increase ROS production via a mitochondria-dependent mechanism in a hepatocarcinoma cell line [38]. In contrast to the present study, this was associated with cytotoxicity within 12–24 hours but with no effects on cell cycle within this time frame. However, previous reports of fucoidan-induced cell death in cancer cells typically used significantly higher concentrations of fucoidan compared to the present study [13,30,38]. We therefore cannot exclude that significantly higher *UPF* concentrations would have also induced cell death in our experimental setting. Given the uncertainty of achievable fucoidan concentrations in the cancer microenvironment in vivo, it is relatively futile to interpret these cytotoxic dose-differences with regards to any therapeutic usefulness. Nevertheless, a mitochondrial mode of action of *UPF* represents an intriguing possibility to explain the cell type specific differences observed in the present study. Given that cancer cells use their mitochondria in a manner very distinct to normal non-immortalized cells [39] future studies will have to delineate the detailed mechanisms of how fucoidans can specifically alter mitochondrial function in cancer cells to induce the cell type-specific effects observed in this and previous studies.

It has to be noted that one of the difficulties of comparative analysis is that fucoidan extracts from different source algae, prepared by using different methods may contain impurities such as polyphenolics, alginates, or other co-extracts. Since the concentrations used in many previous experiments have been relatively high, it cannot be excluded that minor co-extracts rather than the fucoidan itself could have affected the results. Therefore, the current study aimed to use fucoidan with a well-characterized molecular composition and at concentrations low enough to mimic potential physiologically achievable levels. This approach was chosen to enable comparative baseline data for future investigations. In addition, this study used an unbiased approach to identify *Undaria pinnatifida* fucoidan-gene interactions that were successfully translated to mammalian cell responses, which validates the use of *S. cerevisiae* as an initial screening tool. Therefore, this approach could be used in future studies to characterize and more importantly compare different fucoidan preparations that differ in their extraction methods, geographic sources, and species origins. This approach would enable cost effective and reliable comparisons of different extracts and their associated bioactivities.

## 4. Materials and Methods 

### 4.1. Materials

If not otherwise stated, all chemicals were obtained from Sigma-Aldrich (Castle Hill, NSW, Australia). *Undaria pinnatifida* fucoidan (*UPF*) was obtained from Marinova (Cambridge, TAS, Australia). This material was provided with a quoted fucoidan purity of 85.1% (dry weight). The calculation of fucoidan purity requires several inputs that are determined using spectrophotometric assays. The total carbohydrate content of a hydrolyzed sample was determined using the phenol-sulfuric method of Dubois [40,41], while the uronic acid content was determined by spectrophotometric analysis of the hydrolyzed compound in the presence of 3-phenylphenol, based on a method described by Filisetti-Cozzi and Carpita [41]. Sulfate content was analyzed spectrophotometrically using a BaSO_4_ precipitation method (BaCl_2_ in gelatin), based on the work of Dodgson [42], and found to be 24.6%. The molecular weight profile was determined via gel permeation chromatography using a size-exclusion column and reported relative to Dextran standards, with peak molecular weight found at 47.7 kDa.

### 4.2. Yeast Gene Deletion Library

Briefly, a stock concentration (5 mg/mL) of *UPF* was prepared in YPD liquid medium (10 g yeast extract, 20 g peptone, and 20 g dextrose / liter) and sterilized by filtration (cellulose acetate membrane, 0.45 mm pore size, Microscience, Taren Point NSW, Australia). Stock solutions were kept at 4 °C until required. The wild-type (parental) *Saccharomyces cerevisiae* strain BY4741 (MATa *his3*Δ1 *leu2*Δ0 *met15*Δ0 *ura3*Δ0), along with its mutant derivatives covering ~96% of the yeast genome, were obtained from GE Healthcare Dharmacon (Millennium Science, Mulgrave, VIC, Australia). The 4733 gene deletion strains (representing the entire set of nonessential genes for this organism) were supplied frozen in a 96-well microtitre plate format, with individual wells containing settled yeast cells in YPD liquid medium supplemented with 200 µg/mL G418 and 12.5% glycerol. Sub-master plates (Thermo-Fisher Scientific, North Ryde, NSW, Australia) were prepared by thawing the relevant master plate and transferring 10 µl aliquots of each well to 96-well plates containing 90 mL of YPD liquid medium supplemented with G418 (200 µg/mL). The plates were incubated for two days at 30 °C, before 12.5% glycerol was added to each well and the plates stored at minus 80 °C. To prepare experimental 96-well plates containing the deletion strains, the sub-master plates were thawed before a 10 µl aliquot of each well (containing ~1 × 10^6^ cells) was transferred to 96-well plates containing either 90 mL YPD-alone, or 90 mL YPD+500 µg/mL *UPF*. The wild-type (parental) *S. cerevisiae* strain BY4741 was grown overnight at 30 °C in YPD liquid medium to a density of ~1 × 10^8^, before a 10 µl aliquot was inoculated in wells of a 96-well plate containing either 90 mL YPD-alone, or else 90 mLYPD+500 µg/mL *UPF*. All plates were incubated at 30 °C, 5% CO_2_, for 24 h before the growth of the wild-type control and deletion strains was determined by optical density (OD) at 600 nm using a Spectramax M^2^ microplate spectrophotometer (Molecular Devices, San Jose, CA, USA). The sensitivity of each strain to the fucoidan treatment was determined by comparing its growth (optical density) in the absence and presence of fucoidan. Gene deletion strains showing a growth deficit of at least ~1.5 fold were scored as sensitive.

### 4.3. Pathway Analysis

Pathway analysis was performed using String software 10.5 (https://string-db.org) using largely default settings with an interaction score of 0.9 and gene sets restricted to *S.*
*cerevisiae*. Images were generated by hiding disconnected nodes and only showing connected genes. Line thickness was used to illustrate evidence. Settings for the maximum number of interactors to show were selected as first shell: none (only query genes); second shell: none (only query genes). Images were exported from String and pathways were annotated manually in PowerPoint.

### 4.4. Colony Formation Assay

*UPF*-containing media was filtered through 0.45 µm syringe top filters to generate stock solutions of 1 mg/mL. Human colorectal carcinoma cells (HCT-116, 91091055) or non-immortalized human dermal fibroblasts (HDF) (106-05A) were used. Single cell suspensions were seeded in culture media (HCT-116: McCoy’s 5A, M4892) with 2 or 10% FBS, HDF: DMEM (D5523)with 2 or 10% FBS. HCT-116 cells were seeded at 2000 cells per 10 cm culture dish in 2% FCS or 360 cells in 4cm culture dish in 10% while, HDF were seeded at 300 cells per 10 cm culture dish in 10% FCS. After overnight adhesion, the cells were exposed to fucoidan concentrations up to 100 µg/mL for two weeks without media change. The assay was terminated by fixation with 2% *w*/*v* paraformaldehyde in PBS for 10–15 min at RT. Colonies were then stained with 0.25% Coomassie Brilliant Blue in 50% (*v*/*v*) methanol, 10% (*v*/*v*) acetic acid for 5 min at RT. Colonies consisting of more than 50 cells were counted by eye (HCT-116) or under the microscope (HDF). Results are derived from at least four parallel replicates and presented as % colony formation (compared to the untreated control cells). 

### 4.5. WST-1 Assay

This viability assay is based on the enzymatic cleavage of the tetrazolium salt WST-1 to formazan by cellular dehydrogenases present in viable cells. The dye cannot permeate the cell membrane and is reduced outside the cell via the plasma membrane electron transport system. HCT-116 cells were seeded at a density of 5000 cells per well in 96-well plates in 100 µl of 2 or 10% FBS containing McCoy’s media and allowed to adhere overnight. Cells were subsequently treated with various *UPF* concentrations for 24 h to assess acute toxicity. After this time-period the media was removed and 100 µl of fresh media containing 5 µl of WST-1 reagent (Cayman, Sapphire Biosciences Redfern, NSW, Australia) added in each well. Following a 2 h incubation at 37 °C the plate was read in a microplate reader at 450 nm. The WST-1 data for each well were standardized on protein content and the results expressed as % viability.

### 4.6. Detection of DNA Damage by γH_2_AX

To assess if *UPF* causes DNA damage, induction of nuclear γH_2_AX foci was quantified as described previously [43]. Briefly, 1 × 10^5^ HCT-166 or HDF cells were seeded on sterile coverslips in 12 well plates. After adherence, cells were treated with 100 µg/mL fucoidan and incubated for 24 h. Cells were then fixed with 4% PFA in PBS for 15 min and lysed with 1 mL/well of B1 buffer (10% FBS, 0.5% Triton X-100 and PBS) for 1 h at RT on a plate shaker at 200 rpm, followed by two washes with 1 mL of the same buffer. Cells were then incubated with the anti-γH_2_AX antibody (anti-phospho-histone H2A.X(Ser139), (clone JBW301, Lot 2476967, Merck, Bayswater, VIC, Australia) in buffer B1 at 4 °C over night in a humidified chamber. Subsequently, the coverslips were washed 3 x 2 min with W1 buffer (5% FBS, 0.5% Triton X-100 and 0.5× PBS), followed by incubation with species-specific secondary antibody (Alexa Fluor 488 (ab’) 2 fragment of goat anti-mouse IgG (H+L), (ab150117, Abcam, Melbourne, VIC, Australia) for 1 h at RT in B1 buffer. After three washes with W1 buffer and one wash with 1 mL PBS/well, cells were incubated with DAPI (1:10,000) for 5 min to stain the nuclei. After three more washes with 0.5 × PBS, auto-fluorescence was quenched by incubating the cells with 1 mL of W2 buffer (5mM CuSO_4_ in 5 mM ammonium acetate buffer adjusted to pH 5) /well for 20 min at RT. Coverslips were then mounted on microscope slides using Slow Fade Gold anti-fade reagent (S36936, Thermo-Fisher Scientific, North Ryde, NSW, Australia) and stored in the dark until used. Slides were then assessed by counting 3 x 100 cells per condition. The number of cells that contained foci as well as how many foci per cell were present was assessed.

### 4.7. Cell Cycle Analysis

After incubation with 0.1 mg/mL *UPF* for up to 72h HCT-116 cells were harvested by trypsinization and washed with Dulbecco’s phosphate buffered saline (DPBS). 1.5 × 10^6^ cells were resuspended in 750 µl of cold DPBS and left on ice for 10 min. 3 mL of 95% ethanol (−20 °C) were added steadily to the cells, while gently vortexing, before the cells were further fixed for 30 min at 4 °C on ice. Cells were then washed with DPBS twice by centrifuging for 5 min at 850 g. Supernatant was aspired carefully before cells being resuspended in 450 µl of cold DPBS and counted to adjust cell density to 1 × 10^6^ cells/mL. Cells were additionally treated with 5 µl of ribonuclease A (10 mg/mL) before DNA was stained with 100 µl of propidium iodide (PI) (1 mg/ mL) and incubated light-protected at 37 °C for 15 min. Fluorescence was measured using an attune acoustic focusing cytometer (Applied Biosystems, Thermo-Fisher Scientific, North Ryde, NSW, Australia). 10,000 events per sample were collected using the accompanying attune acoustic focusing software. Changes in the percentage of cell distribution for each phase of the cell cycle were used to determine cell cycle effects of *Undaria pinnatifida* fucoidan compared to untreated control cells. Cell cycle data was analyzed using FLOWJO software (version 10, Treestar, Ashland, Oregon, USA).

## 5. Conclusions 

Exposure of a *S. cerevisiae* deletion library to an *Undaria pinnatifida* fucoidan extract (*UPF*) identified a total of 292 genes, whose products are potentially involved in the cellular response to *UPF*. Pathway analysis grouped these genes into a large number of pathways and cellular functions that included ribosome function and biogenesis, cell cycle signaling and DNA repair, nucleotide and amino acid biosynthesis, peroxisomal biosynthesis, mitochondrial function and energy metabolism, RNA synthesis, protein synthesis, and transport. Two exemplary pathways were confirmed in the human colon cancer cell line HCT-116. In this cell line *UPF* reduced colony formation, induced a slow G1 arrest and significant amounts of DNA damage, which was not observed in non-immortalized primary human dermal fibroblasts.

## Figures and Tables

**Figure 1 marinedrugs-17-00054-f001:**
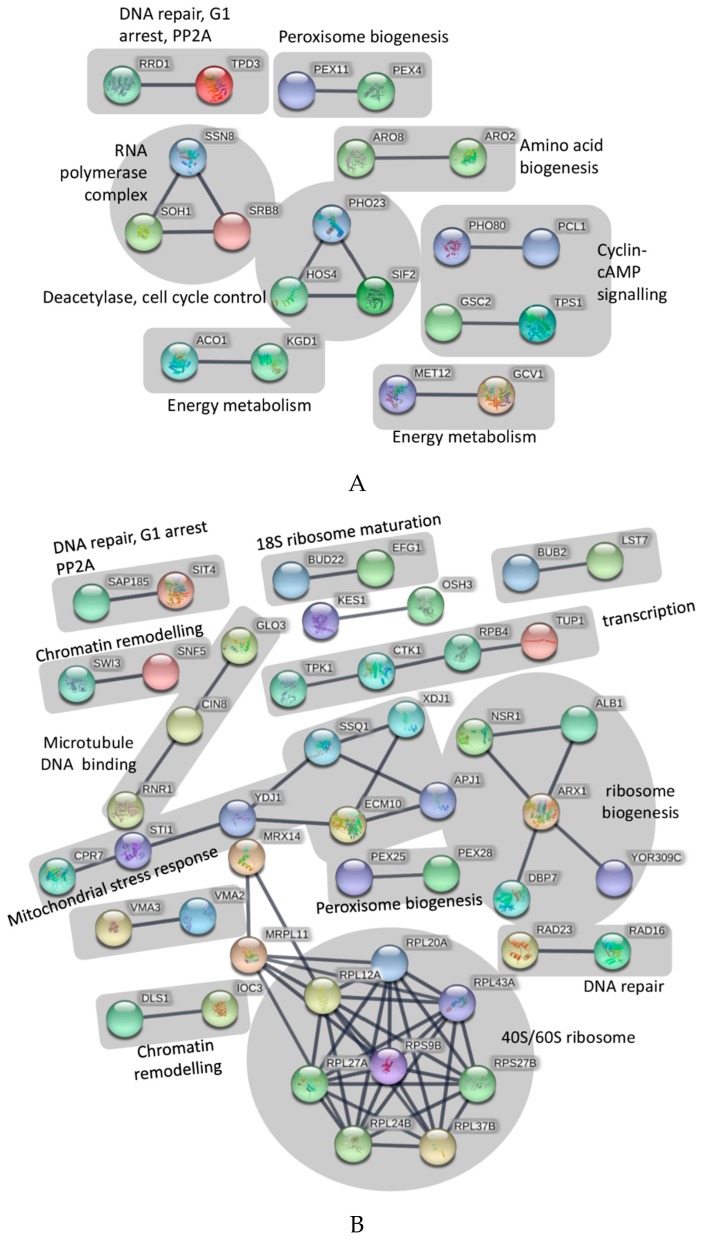
(**A**) Pathway analysis of identified genes that resulted in reduced growth of *S. cerevisiae* in the presence of *UPF*. Pathway analysis was performed with String software (Version 10.5) using an interaction score of 0.9 (‘highest confidence’). The figure only shows connected genes with disconnected nodes hidden. (**B**) Pathway analysis of identified genes that resulted in increased growth of *S. cerevisiae* in the presence of *UPF*. Pathway analysis was performed with String software (Version 10.5) using an interaction score of 0.9 (‘highest confidence’). The figure only shows connected genes with disconnected nodes hidden. (**C**) Pathway analysis of all identified genes that interacted with *UPF*. Pathway analysis was performed with String software (Version 10.5) using an interaction score of 0.9 (‘highest confidence’). The figure only shows connected genes with disconnected nodes hidden.

**Figure 2 marinedrugs-17-00054-f002:**
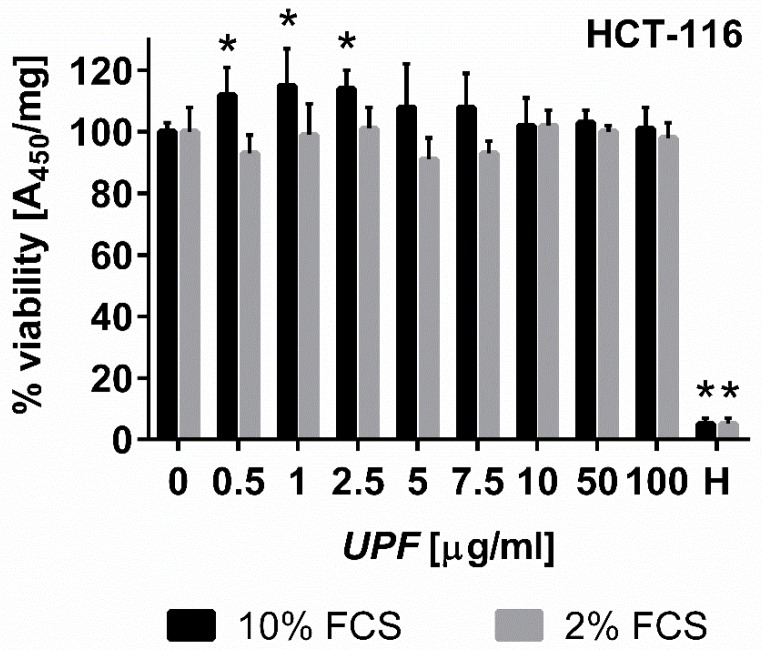
Effect of *UPF* on metabolic activity/viability. Human colon carcinoma cells (HCT-116) were exposed to *UPF* concentrations of up to 100 µg/mL for 24 h before viability was assessed using WST-1 reagent. Data represent one typical experiment out of up to four independent experiments. WST-1 absorption data were standardized on protein content for each well, represent the mean of six individual wells per experiment and are expressed as % viability compared to the untreated control cells. Error bars represent SD with *: *p* < 0.05. Hydrogen peroxide (H, 100 µM) was used as a positive control for toxicity.

**Figure 3 marinedrugs-17-00054-f003:**
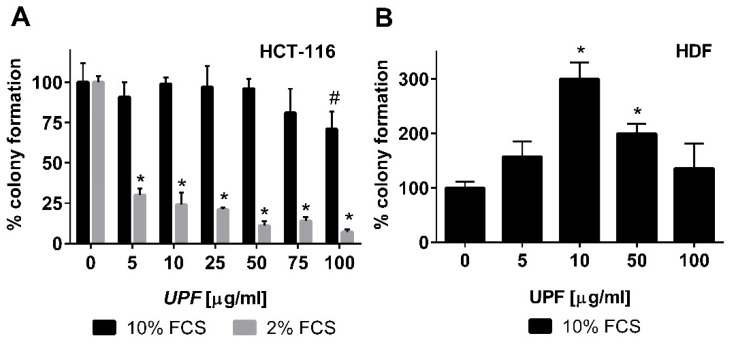
Effect of *UPF* on colony formation. Human colon carcinoma cells (HCT-116, **A**) and human non-immortalized dermal fibroblasts (HDF, **B**) were exposed to *UPF* concentrations up to 100 µg/mL in a colony formation assay. Data represent one typical experiment out of up to four independent experiments. Data is expressed as the mean +/− SD of four plates and expressed as % colony formation compared to the untreated control cells. Error bars represent SD with #: *p* < 0.05 and * *p* < 0.001.

**Figure 4 marinedrugs-17-00054-f004:**
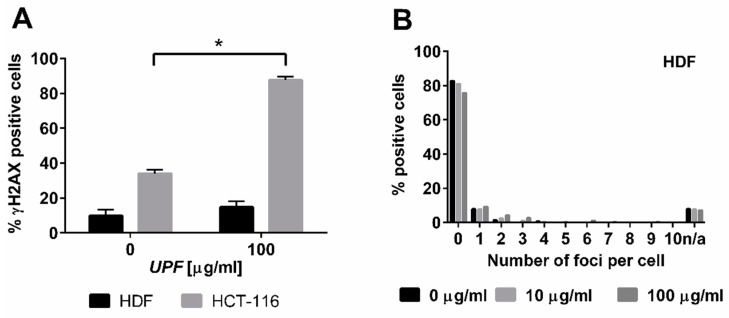
DNA damage induction by *UPF*. Human colon carcinoma cells (HCT-116, **A**, **C**) and human non-immortalized dermal fibroblasts (HDF, **A**,**B**) were exposed to 100 µg/mL *UPF* before γH_2_AX immunostaining was performed (**A**) Data represents the mean of 3 experiments and expressed as % γH_2_AX-positive cells. Error bars represent SD with *: *p* < 0.001. (**B**) Data represent one typical experiment out of three experiments for HDF. n/a: cells excluded from analysis due to unquantifiable staining pattern. (**C**) Representative images for *UPF*-induced induction of nuclear γH_2_AX foci in HCT-116 cells. H_2_O_2_-treatment (100 µM, 30 min) was used as a positive control. Merged images represent software generated false color overlays of DAPI and γH_2_AX signals.

**Figure 5 marinedrugs-17-00054-f005:**
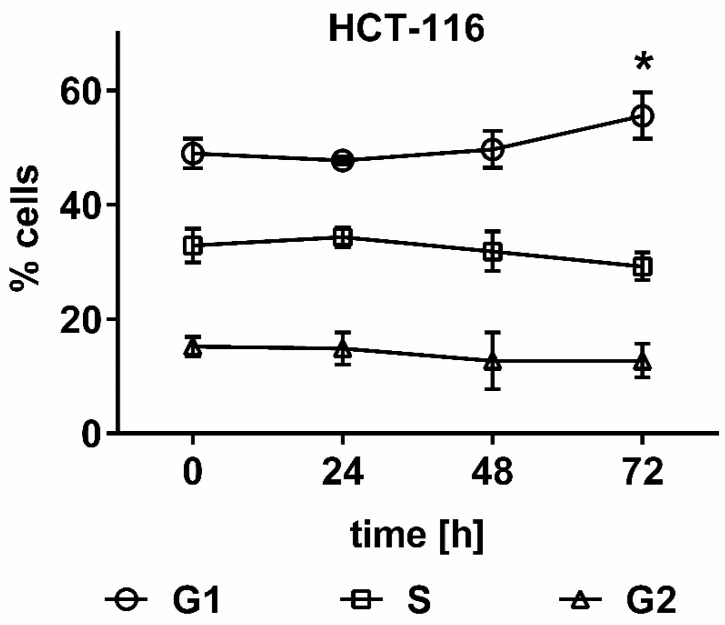
Effect of *UPF* on cell cycle distribution. HCT 116 cells were treated with 100 µg/mL *UPF* for up to 72 h and cell cycle distribution was assessed by flow cytometry. Data represents the mean of four independent experiments performed over a two-month time period. Error bars represent SD with *: *p* < 0.01.

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
