# Peer review of "Pathway Analysis of Fucoidan Activity Using a Yeast Gene Deletion Library Screen"

_marinedrugs, 2019, doi:10.3390/md17010054_

Round 1
Reviewer 1 Report
In my opinion the paper is written properly and I recommend it for publication. Nevertheless some minor corrections must be entered:
· Please correct the e-mail address saved in the affiliate.
· Line 50: If the activity was investigated on other lines then please specify which ones.
· Lines 85, 140, 188, 213, 323 325, 389, 395, 396 and figures 2: There are “ml” and should be mL
· Please standardize the font size in all figures and signatures.
Author Response
1. Please correct the e-mail address saved in the affiliate.
Email address has been corrected as suggested
2. Line 50: If the activity was investigated on other lines then please specify which ones.
We agree with the referee and have added this information into the document (line 50)
3. Lines 85, 140, 188, 213, 323 325, 389, 395, 396 and figures 2: There are “ml” and should be mL
As suggested by the reviewer, all occurrences of “ml” in the manuscript have been changed to “mL”
4. Please standardize the font size in all figures and signatures.
We attempted to standardize font size in all figures as much as possible but the final publishing step will likely change the reproduction size of our figures to a similar size. This will invariably change font sizes again. Nevertheless, for all figures we attempted to align font sizes as closely as possible based on the assumption of equal-sized reproduction of individual graphs by the journal.
Reviewer 2 Report
1- In page 6, line 159, there is indication of 2 weeks of continuous treatment, would that long of incubation gives accurate results? wont this rise a chance of false positive results?
2- Why the authors decided to test the activity on only one cancer cell line? the justification for their choice of the cell line is missing.
3- In fig2 and fig3 why the authors decided to use different concentration of UPE on the same cell line while they comparing the effects of 2% and 10% of FCS? this need to be presented in better way for the reader.
4- The sentences in lines 183 &184 need to re-phrase it, as each cell line response to the antiproliferation agent differently under the lab condition.
5- Fig5, line 212 the figure legend need to be corrected.
6- Line 271 mentioned the effect on ROS while there is nothing in the results demonstrated/ showed this. This need to be added to the results or otherwise delete the sentence.
7- Lines 203 & 204 speak about the well known source of fucoiden, and as mentioned that has been provided commercially, so why the authors didnt show the concentration in whole experiments with mmol or micro mol instead of micrograms?
Author Response
1. In page 6, line 159, there is indication of 2 weeks of continuous treatment, would that long of incubation gives accurate results? wont this rise a chance of false positive results?
The colony formation assay is a highly standardized assay, commonly used to measure drug toxicities. It is seen as the gold standard assay for long term toxicity assessment. There are two variants of this assay. The first one, used by us, measures long term toxicity in terms of proliferative capacity of cells in the continued presence of the test compound. This form of the assay is the most widely used. The second variant of the assay only exposes cells temporarily to the test compound and then lets the cells form colonies in the absence of test compound. This assay variant is used to measure the repair/adaptation capacity of cells to a toxin. This means the cells capacity to temporarily deal with a potential toxin is tested. In contrast the assay version we have used detects intrinsic compound toxicity over extended periods of time. This was the question we wanted to answer in this study since fucoidan is typically used over longer periods of time.
2. Why the authors decided to test the activity on only one cancer cell line? the justification for their choice of the cell line is missing.
We agree that the rationale to test the activity of UPF on only one cell line might not have been clear enough and we have now amended the manuscript to address this concern (lines 131-134).
3. In fig2 and fig3 why the authors decided to use different concentration of UPE on the same cell line while they comparing the effects of 2% and 10% of FCS? this need to be presented in better way for the reader.
We agree with the referee and have revised the text to clarify this point in the results section (lines 160-166).
4. The sentences in lines 183 &184 need to re-phrase it, as each cell line response to the antiproliferation agent differently under the lab condition.
We agree with the reviewer that the HCT-116 cell line responded very differently to the primary human skin cell strain HDF in response to UPF exposure and the description of our results was possibly not detailed enough. We have therefore reworded this section to highlight the important difference between the two cell types (lines 191-199).
5. Fig5, line 212 the figure legend need to be corrected.
We assume the reviewer refers to the font size of the figure legend that needs to be corrected and have therefore checked and adjusted the font size of all figure legends if necessary.
6. Line 271 mentioned the effect on ROS while there is nothing in the results demonstrated/ showed this. This need to be added to the results or otherwise delete the sentence.
We agree with the reviewer that we did not show ROS production in our study. However, fucoidan-induced ROS was shown in two previous studies (36,37) that we cite in the discussion of our manuscript in the sentence before. Therefore, the production of ROS by UPF is a possibility and discussed as such by us. Therefore, this sentence does not have to be deleted.
7. Lines 203 & 204 speak about the well known source of fucoiden, and as mentioned that has been provided commercially, so why the authors didnt show the concentration in whole experiments with mmol or micro mol instead of micrograms?
Similar to Heparin, Fucoidan is not a single molecule but a complex mixture of polysaccharides of different length and structure. Therefore, unlike applicable to a purified single well-defined molecule, fucoidan concentrations are generally given as mg/ml (as it is also done for Heparin).